# BACE: Behavior-Adaptive Connectivity Estimation for Interpretable Graphs of Neural Dynamics

## Abstract

Understanding how distributed brain regions coordinate to produce behavior requires models that are both predictive and interpretable. We introduce Behavior-Adaptive Connectivity Estimation (BACE), an end-to-end framework that learns context-specific, directed inter-regional connectivity directly from multi-region intracranial local field potentials (LFP). BACE aggregates many micro-contacts within each anatomical region via per-region temporal encoders, applies a learnable adjacency specific to each behavioral contex, and is trained on a forecasting objective. On synthetic multivariate time series with known graphs, BACE accurately recovers ground-truth directed interactions while achieving forecasting performance comparable to state-of-the-art baselines. Applied to human subcortical LFP recorded simultaneously from eight regions during a cued reaching task, BACE yields an explicit $8 \times 8$ connectivity matrix for each within-trial behavioral contex. The resulting behavioral contex-specific graphs reveal behavior-aligned reconfiguration of inter-regional influence and provide compact, interpretable adjacency matrices for comparing network organization across behavioral conte. By linking predictive success to explicit connectivity estimates, BACE offers a practical tool for generating data-driven hypotheses about the dynamic coordination of subcortical regions during behavior.

## 1 Introduction

Understanding how patterns of interaction between neural populations reorganize with behavior is central to systems neuroscience and to applications that decode or modulate brain activity. These interactions are often framed under the umbrella of brain connectivity, distinguished as structural (anatomical pathways), functional (statistical dependencies), and effective (directed influence)—complementary lenses on how distributed circuits communicate and adapt to task demands (Friston, 2011; Fornito et al., 2016). Reviews emphasize both the promise and the challenges of connectivity-centric approaches, particularly the need to capture short-timescale dynamics while maintaining interpretability (Mohammadi & Karwowski, 2025; Srivastava et al., 2022).

Graph-based formulations make this agenda concrete: representing brain regions as nodes and their relationships as edges enables quantitative analysis of modularity, hubs, and task-dependent reconfiguration (Bassett & Sporns, 2017; Cole et al., 2013). Such network perspectives motivate methods that go beyond static, correlation-only descriptions toward dynamic, directed estimates that better align with mechanistic questions.

We focus on intracranial local field potentials (LFPs), a high-temporal-resolution measure of population activity recorded simultaneously from multiple deep-brain regions (Buzsáki et al., 2012; Einevoll et al., 2013; Anonymized, Anonymized). Multi-region LFP imposes unique modeling requirements: (i) many micro-contacts per region must be consolidated into coherent region-level trajectories; (ii) neural activity reconfigures across well-defined behavioral contexts; and (iii) the spatial layout of contacts is high-dimensional and non-Euclidean (Bronstein et al., 2021). Existing pipelines often average across behavior with a single correlation-based graph (Fornito et al., 2016) or impose hard-coded anatomical connectivity (Hutchison et al., 2013; Preti et al., 2017), limiting their ability to capture context-specific, directed interactions.

Recent machine learning advances in graph learning from neural time series highlight the opportunity but do not yet fill this gap. Models such as GraphS4mer couple sequence modeling with learned graphs but are primarily evaluated on decoding tasks without exposing directed region-level connectivity (Tang et al., 2023). In fMRI, architectures like STAGIN improve classification via spatio-temporal attention but operate on undirected functional connectivity (Kim et al., 2021a). Forecasting-based models such as AMAG learn adaptive graphs from cortical LFP, yet they focus on channel-level grids and are not designed for multi-region, behavior-conditioned settings (Li et al., 2023). A framework that yields compact, directed, context-specific connectivity estimates for invasive multi-region recordings remains lacking.

We introduce **BACE** (Behavior-Adaptive Connectivity Estimation), an encoder–projector–decoder framework that learns directed, behavioral context-specific effective connectivity from multi-region LFP. BACE (i) aggregates many contacts within each anatomical region via per-region temporal encoders; (ii) learns a distinct adjacency for each behavioral context, with a parameterization that separates edge pattern ("who influences whom") from broadcast strength (per-row gains); and (iii) couples the learned graphs to signal dynamics through a forecasting objective with continuity and smoothness priors.

On synthetic multivariate time series with known ground-truth graphs, BACE accurately recovers directed interactions. Applied to human subcortical LFPs recorded from eight regions during a cued reaching task, BACE produces explicit $8 \times 8$ connectivity matrices for each behavioral segment. Beyond predictive accuracy, we assess reproducibility, quantify uncertainty, and test across-contexts differences, turning learned adjacencies into auditable scientific objects rather than opaque model internals. We summarize the main contributions of our work below:

- **A region-level graph model for neural forecasting.** BACE couples per-region temporal encoders with a learnable adjacency to capture directed inter-regional influence.

- **Recovery of interaction structure.** On synthetic datasets with known ground truth, BACE reliably reconstructs directed connectivity using only a forecasting signal.

- **Behavior-conditioned graphs.** BACE yields compact, signed inter-regional adjacencies aligned to behavioral contexts, enabling direct comparison of task-specific network organization.

- **Interpretability.** An adjacency parameterization decouples edge pattern from transmission strength, producing small, explicit matrices suitable for neuroscientific analysis.

## 2 RELATED WORKS

Our work is situated at the intersection of four key areas: (i) forecasting-based modeling of neural dynamics, (ii) dynamic brain-graph learning, (iii) graph-structure learning from time series, and (iv) interpretability in graph-based neuroscience. While these fields have advanced rapidly, methods that learn compact, directed, and behavior-specific inter-regional graphs remain underexplored.

**Modeling neural dynamics with forecasting.** Modeling neural dynamics has evolved from linear state–space and Gaussian process factor models (Dahlhaus, 2000; Yu et al., 2008; Gao et al., 2016) to modern deep learning approaches. To capture complex nonlinearities, the field has widely adopted recurrent networks (Hochreiter & Schmidhuber, 1997; Pandarinath et al., 2018), variational autoencoders (Zhou & Wei, 2020), neural ODEs (Kim et al., 2021b), and Transformers adapted to neural time series (Ye & Pandarinath, 2021; Le & Shlizerman, 2022). A common thread across these modern approaches is the use of predictive objectives, which have proven effective for learning latent representations aligned with behavior (Sani et al., 2021). Despite their success in uncovering latent dynamics, these methods are generally not designed to yield the explicit, interpretable, and behavior-conditioned connectivity graphs that are the primary focus of this work

**Dynamic brain-graph learning.** Graph neural networks have emerged as a powerful framework for modeling time-resolved brain activity. In fMRI, spatiotemporal attention and Transformer-based architectures such as STAGIN (Kim et al., 2021a) learn dynamic embeddings of functional connectivity for tasks like cognitive-state or demographic classification. Structured state-space hybrids, including GraphS4mer (Tang et al., 2023) and more recent Mamba-based variants (Behrouz &

Hashemi, 2024), further extend this direction by coupling long-range temporal encoders with adaptive graph learners. Collectively, these works demonstrate the utility of spatiotemporal GNNs across neuroimaging modalities. However, they are generally evaluated on EEG or fMRI classification benchmarks and yield latent embeddings or undirected functional connectivity, rather than compact, directed adjacency matrices explicitly conditioned on behavioral context.

**Graph-structure learning from time series.** A distinct but related line of work, Graph-Structure Learning (GSL), aims to infer latent edges jointly with temporal dynamics (Jin et al., 2020). This is particularly relevant for neuroscience, where the true underlying graph is unknown. AMAG exemplifies this direction by introducing additive and multiplicative message passing and, importantly, learning a trainable channel-level adjacency jointly with forecasting on cortical LFP and synthetic benchmarks (Li et al., 2023). The broader GSL literature includes variational models for discovering latent graphs (Huang & Yu, 2022) and bilinear graph learners for edge inference (Zhu et al., 2020). These studies validate the principle that predictive objectives can successfully uncover interaction structures from time-series data. However, they typically operate at the fine-grained channel level and do not explicitly model how such structure reconfigures with behavior. BACE addresses this gap by learning *directed, region-level* graphs explicitly conditioned on behavioral context, enabling direct and interpretable comparisons of network organization across moments within a trial.

**Interpretability in graph-based neuroscience.** Interpretability efforts in brain GNNs span ROI saliency and causal subgraph discovery. For example, BrainGNN highlights salient regions through ROI-aware pooling and regularizers, but it does not produce an explicit connectivity matrix (Li et al., 2021). CI-GNN learns sparse, causally-relevant subgraph masks from static, undirected fMRI connectivity graphs, linking them to diagnostic predictions (Zheng et al., 2024). Post-hoc explainers such as GNNExplainer (Ying et al., 2019) and domain-specific extensions like BrainNNExplainer target edge-level saliency, but again operate in settings focused on classification. Benchmarking and survey efforts such as NeuroGraph (Said et al., 2023) and recent reviews of brain GNNs (Bessadok et al., 2022) confirm that most interpretability work remains tied to fMRI decoding tasks and post-hoc analysis. In contrast, BACE directly learns inter-regional graphs from invasive LFP dynamics and conditions them on behavioral context, producing compact, directed adjacency matrices at the region level designed as primary scientific objects for comparing network structure across behavioral contexts.

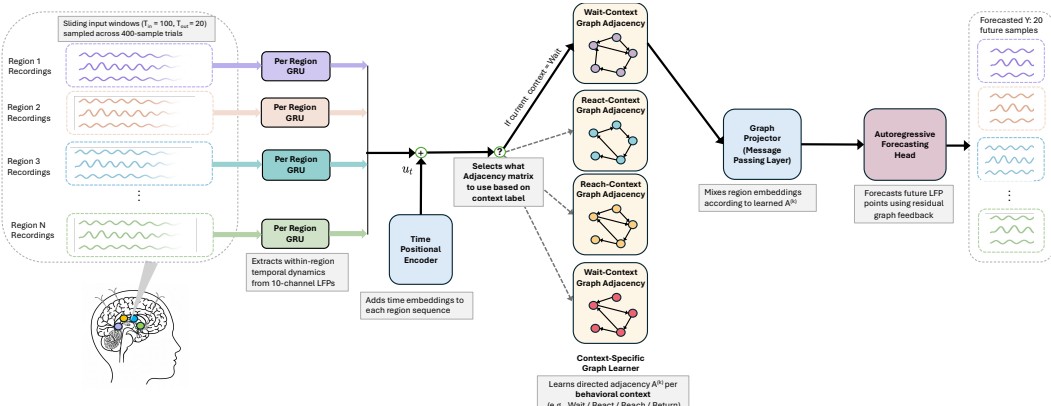

Figure 1: **BACE architecture.** Per-region temporal encoders (independent GRUs) extract local dynamics from raw neural windows. A behavioral-specific graph learner produces one directed adjacency $\mathbf{A}_\phi$ for each behavioral segment, which the graph projector applies to mix regional features. The forecaster head then predicts the unobserved future window using a residual objective. This end-to-end design links forecasting directly to explicit, behavioral-specific connectivity estimates.

## 3 METHODS

We aim to learn dynamic effective connectivity between neural regions from multivariate time series. Formally, let $\mathbf{X}_{t-T_{\text{in}}+1:t} \in \mathbb{R}^{N \times C \times T_{\text{in}}}$ denote neural activity from $N$ regions, each with $C$ channels, over a past window of length $T_{\text{in}}$. Our objective is to learn a function

$$f_\theta : \left( \mathbf{X}_{t-T_{\text{in}}+1:t},\ k \right) \ \mapsto \ \left( \hat{\mathbf{Y}}_{t+1:t+T_{\text{out}}},\ \mathbf{A}^{(k)} \right).$$

that predicts the next $T_{\text{out}}$ time steps while simultaneously identifying a directed connectivity matrix $\mathbf{A}^{(k)} \in \mathbb{R}^{N \times N}$ associated with behavioral context $k$.

The problem is thus framed as joint forecasting and connectivity estimation. The key principle is that if a connectivity pattern is explanatory, it is expected to improve prediction of future neural dynamics. This allows us to cast connectivity estimation as a supervised learning problem grounded in predictive modeling.

Our model comprises four components:

**Regional Encoder.** In our dataset, each region contains multiple channels of neural recordings that capture locally correlated activity. To encode these within-region dynamics, we employ a set of **regional Gated Recurrent Unit (GRU) encoders** (Cho et al., 2014), one per region:

$$\mathbf{H}_r = \text{GRU}_r(\mathbf{X}_r), \quad r = 1, \ldots, N,$$

where $\mathbf{X}_r \in \mathbb{R}^{C \times T_{\text{in}}}$ is the multichannel input for region $r$.

This design models local dynamics independently, preserving regional separation before cross-regional interactions are introduced. This is essential, as inter-regional influences are explicitly represented in the learned connectivity matrix later in the model. Encodings are further stabilized with layer normalization. After encoding, a lightweight positional embedding is concatenated to each timestep to provide explicit information about relative time within the input window.

**Context-Specific Graph Learner.** Our Context-Specific Graph Learner produces a directed graph per behavioral context. For each behavioral context $k$, we represent the system as a directed graph $G^{(k)} = (V, E^{(k)}, A^{(k)})$, where $V = \{1, \ldots, N\}$ denotes the set of regions, $E^{(k)}$ the set of directed edges, and $A^{(k)} \in \mathbb{R}^{N \times N}$ the weighted adjacency matrix. An entry $A_{ij}^{(k)}$ encodes the directed influence from region $j$ to region $i$.

For each behavioral context $k$, we learn a separate adjacency $A^{(k)}$. Rows are normalized to have unit $\ell_1$ norm, i.e. $\sum_j |A_{ij}^{(k)}| = 1\ \forall i$, ensuring stable and interpretable graphs where each row can be read as a distribution of incoming influences to region $i$. During training, the context label of each trial acts as a selector, routing the input to the corresponding adjacency $A^{(k)}$.

**Graph Projector.** Having defined a context-specific adjacency $A^{(k)}$, the next step is to propagate information between regions. Each region's output from the encoder is a sequence of temporal embeddings $\mathbf{H}_i \in \mathbb{R}^{T \times d}$. To incorporate inter-regional influences, we update these embeddings with two components: (i) a self-update that preserves the region's own trajectory, and (ii) a neighbor-mixing term that aggregates activity from other regions according to $A^{(k)}$.

Formally, for region $i$ and time $t$,

$$\mathbf{Z}_{i,t} \ = \ W_{\text{self}}\, \mathbf{H}_{i,t} \ + \ \sum_j A_{ij}^{(k)}\, W_{\text{neigh}}\, \mathbf{H}_{j,t},$$

where $W_{\text{self}}$ and $W_{\text{neigh}}$ are learnable linear maps. The first term retains the region's own dynamics, while the second injects information from neighbors in proportion to the directed influences encoded in $A^{(k)}$. To ensure stability, the neighbor transformation is spectrally normalized.

This projection can be viewed as a single message-passing layer: each region maintains its encoded dynamics while receiving context-dependent inputs from other regions. The resulting sequence $\mathbf{Z} \in \mathbb{R}^{N \times T \times d}$ forms the input to the forecasting head.

**Autoregressive forecasting head.**    The final stage generates future trajectories autoregressively. Given graph-projected embeddings $\mathbf{Z} \in \mathbb{R}^{N \times T_{\text{in}} \times d}$, the decoder produces a sequence of length $T_{\text{out}}$. The decoder attends to the full encoder sequence of past inputs, with attention weights that adapt across forecast steps to provide context for prediction.

At each forecast step $t$, the most recent prediction $\hat{\mathbf{y}}_{t-1}$ (initialized with the last observed sample) is refined through adjacency mixing:

$$\tilde{\mathbf{y}}_{t-1} = \hat{\mathbf{y}}_{t-1} + \alpha\, A^{(k)} \hat{\mathbf{y}}_{t-1}.$$

This step reintroduces the learned connectivity into the forecasting loop, ensuring that predictions remain explicitly shaped by inter-regional influences rather than relying only on the earlier graph-projected encoder states. The refined feedback $\tilde{\mathbf{y}}_{t-1}$ is then combined with the context vector $\mathbf{c}_t$ from attention to update the decoder state and produce an increment $\Delta\hat{\mathbf{y}}_t$.

Absolute forecasts are recovered recursively as

$$\hat{\mathbf{y}}_t = \hat{\mathbf{y}}_{t-1} + \Delta\hat{\mathbf{y}}_t,$$

a residual formulation that stabilizes training and mitigates drift in multi-step prediction. A linear readout maps regional hidden states back to the original channel space, yielding predictions $\hat{\mathbf{Y}} \in \mathbb{R}^{N \times C \times T_{\text{out}}}$ aligned with the ground-truth future window.

**Training and Optimization**    The model is trained end-to-end to minimize forecasting error over the next $T_{\text{out}}$ samples, with lightweight regularizers that encourage sparsity in the learned graphs and stabilize short-horizon predictions. Optimization is performed with Adam and early stopping based on validation loss. Full loss definitions, initialization, and optimization details are provided in the Appendix.

## 4    EXPERIMENTS AND RESULTS

### 4.1    SYNTHETIC DATA: RECOVERY OF GROUND-TRUTH CONNECTIVITY

We first evaluated BACE on synthetic multivariate time series with known ground–truth connectivity to test whether it can recover the known interaction structure. These datasets were designed only to mimic the organization of our neural recordings (8 regions, hundreds of trials segmented into short windows); details of the real dataset follow in the next section.

**Dataset design.**    We generated two suites of datasets, each containing four regimes $\mathcal{D}_1$–$\mathcal{D}_4$. In the *structured suite*, each regime is governed by a sparse adjacency matrix with identical row degree but differing placement of nonzeros, yielding multiple shifted but visually structured connectivity patterns. In the *stochastic non–Gaussian suite*, datasets were generated using linear non–Gaussian dynamical systems, following prior work on learning structure from dynamics (Li et al., 2023; Biswas & Shlizerman, 2022b;a; Song et al., 2009). The governing dynamics take the form $\mathbf{X}_t = \mathbf{X}_{t-1} + G(\mathbf{X}_{t-1}, \mathbf{A}) + \boldsymbol{\mu}_t$, where $G(\mathbf{X}_{t-1}, \mathbf{A}) = (-\lambda \mathbf{I} + \gamma \mathbf{A})\, \mathbf{X}_{t-1}$ encodes a linear leak term $\lambda > 0$ and adjacency–mediated coupling $\gamma \mathbf{A}$. The noise process $\boldsymbol{\mu}_t$ is autoregressive with uniform innovations, making it colored and non–Gaussian.

**Synthetic validation.**    For synthetic validation we applied the same BACE architecture described in Methods, with no supervision of spatial connectivity. Recovery was assessed with two metrics: (i) F1@$k_{\text{row}}$, which tests whether the top-$k$ entries in each row match the ground-truth nonzeros (with k=2 set to row degree), and (ii) Pearson correlation between learned and ground-truth weights. On the structured suite, BACE attained perfect edge recovery (F1@$k_{\text{row}}$=1.0 for all $\mathcal{D}_1$–$\mathcal{D}4$) with correlations 0.84/0.93/0.89/0.89 (mean 0.89). On the stochastic non-Gaussian suite, recovery remained strong with F1@$k$row = 0.94/1.00/1.00/1.00 (mean 0.98) and correlations 0.88/0.94/0.88/0.90 (mean 0.90). Figure 2 illustrates representative cases: strong edges align with ground truth, and weaker off-diagonal influences are captured with appropriate magnitude. These results show that BACE reliably reconstructs directed adjacency structure while being trained solely for forecasting, supporting its use for effective connectivity estimation.

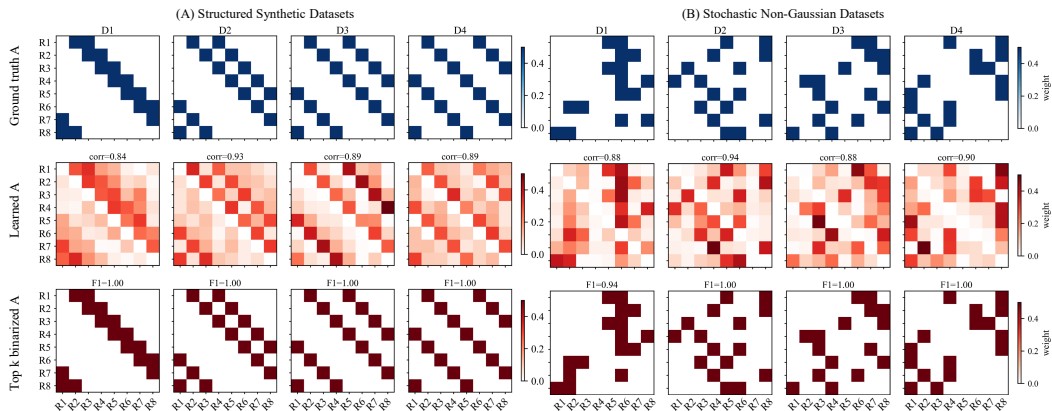

Figure 2: **Adjacency recovery on synthetic datasets.** (**A**) Stochastic non-Gaussian suite; (**B**) structured suite. Columns correspond to the four datasets $\mathcal{D}_1$–$\mathcal{D}_4$. Rows: (*top*) ground-truth adjacency $\mathbf{A}$; (*middle*) BACE's learned adjacency $\hat{\mathbf{A}}$ (row-normalized at inference); (*bottom*) top-$k$ binarization of $\hat{\mathbf{A}}$ with $k$ equal to the ground-truth row degree (here $k=2$). Numbers above each panel indicate F1@$k_{\text{row}}$. Quantitatively, BACE achieves mean correlation 0.89 and mean F1@$k_{\text{row}}=1.00$ on the structured suite, and 0.90 / 0.98 on the stochastic suite.

## 4.2 EVALUATION ON HUMAN LFP DATA

**Dataset and Preprocessing.** We analyzed intracranial local field potentials (LFP) from a pediatric participant with dyskinetic cerebral palsy undergoing deep brain stimulation (DBS) as part of clinical treatment. Neural data were recorded during a cued reaching task. The dataset contained 326 valid trials, each segmented into four behavioral segments: Wait (pre-go cue baseline), React (post-go cue, pre-movement), Reach (forward reach toward target), and Return (movement back to the home button). Each segment was represented by a 400 ms window (400 samples at 1 kHz), yielding consistent trial segments across conditions.

Recordings were obtained from 80 micro-contact channels across eight subcortical regions: internal globus pallidus anterior (GPi-anterior), internal globus pallidus posterior (GPi-posterior), ventral intermediate nucleus of the thalamus (VIM), and subthalamic nucleus (STN) in both hemispheres (10 channels from each region). Preprocessing included a 50 Hz low-pass filter, common average referencing, and downsampling to 1 kHz. For forecasting, each 400-sample segment was further split into sliding windows of 100-sample input ($T_{\text{in}}$) and 20-sample prediction ($T_{\text{out}}$) with stride 20. To prevent leakage, train/validation/test splits were made at the trial level, and all normalization statistics were computed on the training set. Task schematic and behavioral segments are illustrated in Fig. 3; full details of task design and segmentation criteria are provided in the Appendix.

**Benchmark Methods.** To contextualize BACE's performance, we compare against representative models spanning recurrent, Transformer, and graph-based approaches. As a recurrent baseline we include a standard **LSTM**(Hochreiter & Schmidhuber, 1997), a canonical gated architecture widely used for sequential modeling. From the Transformer family, we evaluate the **Neural Data Transformer (NDT)**(Ye & Pandarinath, 2021), designed for neural sequence modeling with masked reconstruction objectives. Among graph-based methods, we consider three widely used variants: (i) **DCRNN**(Li et al., 2017), a diffusion convolutional recurrent network that integrates graph diffusion into GRU updates over a fixed adjacency; (ii) **GWNet**(Wu et al., 2019), a spatio-temporal graph convolutional model that combines dilated causal convolutions with adaptive graph learning for spatial mixing; and (iii) **AMAG**(Li et al., 2023), a GNN framework that adaptively learns dynamic channel-level graphs for neural forecasting.

**Experimental Setup.** All models, including BACE and the five baselines, were trained and evaluated on the same intracranial LFP dataset described above. Each trial was segmented into four behavioral segments of 400 samples, further divided into sliding windows of $T_{\text{in}}=100$ input steps

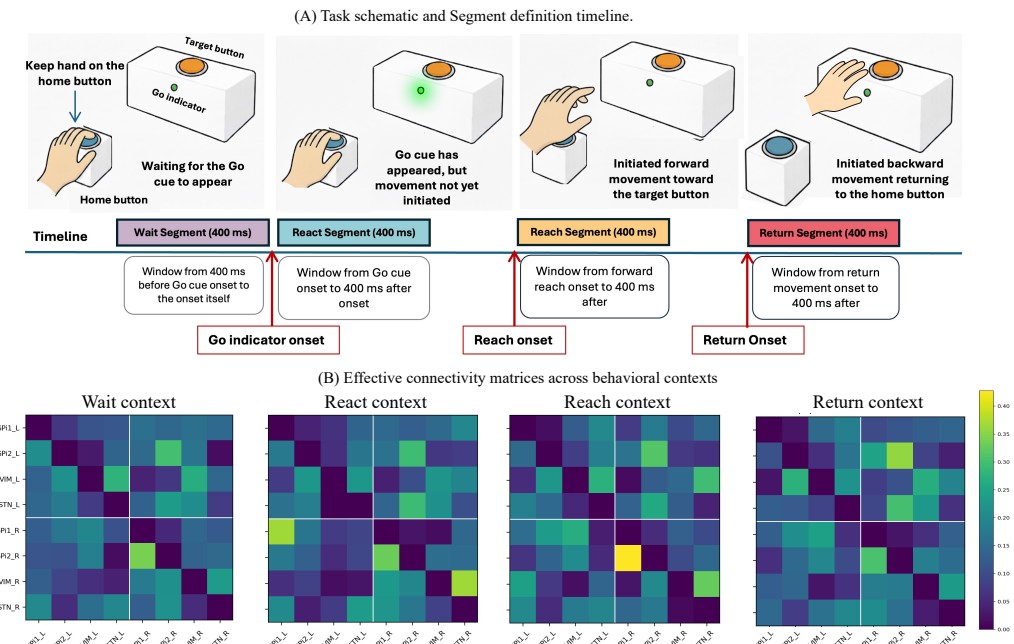

Figure 3: **(A)** Task schematic and behavioral segmentation timeline. Each trial comprised four stages with a 400 ms analysis window: Wait ($-400$ ms to Go cue onset), React (0–400 ms after Go cue onset), Reach (0–400 ms after forward reach onset), and Return (0–400 ms after return onset). **(B)** behavioral-specific $8 \times 8$ effective connectivity matrices estimated by BACE across eight subcortical regions (GPi1, GPi2, VIM, STN; left and right hemispheres). Matrix entry $A_\phi[i, j]$ denotes directed influence from source region $j$ (column) to target region $i$ (row).

and $T_{\text{out}}{=}20$ forecast steps. Models were optimized with the Adam optimizer under their standard hyperparameters (details in Appendix). Training was performed with early stopping on validation loss, and the best checkpoint was used for reporting. Performance was assessed on the held-out test set using three metrics: mean squared error (MSE), coefficient of determination ($R^2$), and Pearson correlation (Corr). MSE measures absolute deviation, $R^2$ quantifies explained variance, and Corr evaluates trend alignment between predicted and ground-truth signals.

**Model Performance.** We evaluate the ability of BACE and other models to forecast future neural activity, with multi-step results summarized in Table 1. Each model was trained and tested on identical splits, and reported values reflect the mean with standard deviation over three runs. Among non-graph methods, LSTM and NDT achieved reasonable accuracy but lagged behind graph-based models, explaining less than 75% of the variance on average, consistent with their limited ability to capture structured inter-regional dependencies. Graph-based methods performed substantially better. DCRNN reached $R^2$ of 0.79 confirming the benefit of incorporating graph priors, but reflecting the limitation that its adjacency is pre-computed and fixed during training. GWNet improved further with adaptive graph learning, and AMAG provided a modest additional gain, validating the utility of dynamic spatial structure. BACE consistently achieved the best performance across all metrics, with an $R^2$ of 0.884 and correlation of 0.94, reflecting both lower forecast error and tighter alignment with the true trajectories. By combining strong predictive accuracy with behavioral-specific graph estimation, BACE offers not only numerical improvement but also interpretable connectivity estimates among brain regions.

**Computation complexity.** We compare the number of trainable parameters #P(M) and peak GPU memory $M(GB)$ in Table 2. Parameter counts are determined by the model architecture and thus provide a stable point of comparison, whereas memory usage and runtime are more sensitive to implementation and hardware. BACE has the smallest footprint overall (0.13M parameters, 0.20 GB memory) while training in ~9.8s per epoch. LSTM and NDT require substantially more parameters

Table 1: Multistep forecasting results on the LFP dataset with benchmark methods and BACE. $\uparrow$ indicates higher is better, $\downarrow$ indicates lower is better.

| Model | $R^2 \uparrow$ | Corr $\uparrow$ | MSE $\downarrow$ |
|---|---|---|---|
| LSTM | $0.7271 \pm 2e^{-3}$ | $0.8353 \pm 4e^{-3}$ | $0.2690 \pm 3e^{-4}$ |
| NDT | $0.7336 \pm 3e^{-3}$ | $0.8610 \pm 3e^{-3}$ | $0.2370 \pm 3e^{-4}$ |
| DCRNN | $0.7895 \pm 2e^{-3}$ | $0.8892 \pm 4e^{-3}$ | $0.1822 \pm 6e^{-4}$ |
| GWNet | $0.8632 \pm 3e^{-3}$ | $0.9296 \pm 4e^{-3}$ | $0.1184 \pm 7e^{-4}$ |
| AMAG | $0.8676 \pm 4e^{-3}$ | $0.9315 \pm 4e^{-4}$ | $0.1146 \pm 4e^{-4}$ |
| BACE | $\mathbf{0.8836} \pm \mathbf{3e^{-3}}$ | $\mathbf{0.9400} \pm \mathbf{3e^{-4}}$ | $\mathbf{0.0990} \pm \mathbf{2e^{-4}}$ |

Table 2: Computation complexity for multi-step forecasting. BACE values are in bold; values better than BACE are also highlighted.

| | LSTM | NDT | GWNet | DCRNN | AMAG | BACE |
|---|---|---|---|---|---|---|
| #P (M) | 0.45 | 0.63 | 0.48 | 0.30 | 0.26 | **0.13** |
| $M$ (GB) | **0.15** | 0.84 | 5.54 | 5.22 | 3.30 | **0.20** |

(0.45M and 0.63M, respectively). GWNet, DCRNN, and AMAG all introduce graph operators that increase both parameter counts and memory consumption, with GWNet the heaviest in terms of parameters. In particular, DCRNN and AMAG have parameter sizes close to BACE but still incur memory cost due to the overhead of diffusion and adaptive graph operators. These results indicate that BACE's gains arise from its architectural design rather than increased capacity, combining efficiency with interpretability. Compute specifications are provided in Appendix B.3.

### 4.3 ANALYSIS OF LEARNED CONNECTIVITY

**Reproducibility.** We first asked whether BACE's learned connectivity graphs are stable across resampled subsets of trials. Split–half analysis showed high reproducibility, with correlations of absolute adjacency magnitudes $|\mathbf{A}^{(k)}|$ exceeding $r > 0.8$ for all behavioral segments, corresponding to Spearman–Brown corrected (Spearman, 1910; Brown, 1910) full–data repeatability above 0.9. These results indicate that BACE yields reliable, segment–specific connectivity estimates suitable for scientific interpretation (see Appendix C for full details).

**Edge reliability and reconfiguration.** We next tested whether directed connectivity patterns reorganize with behavioral context. For each adjacency $\mathbf{A}^{(k)}$, we performed nonparametric bootstrap resampling to estimate confidence intervals and assessed across-context differences with FDR control (Bradley & Tibshirani, 1993). Several edges exhibited significant changes between contexts, as illustrated in Fig. 4A. These findings suggest that subcortical networks are not strictly static but can flexibly adjust their directed interactions in relation to task demands (Cole et al., 2013). Full bootstrap procedures and edgewise statistics are reported in Appendix C.

**Spatial organization.** We next examined whether the learned graphs reflect known spatial regularities of brain connectivity. First, consistent with the principle that networks favor short-range, intra-hemispheric coupling (Bullmore & Sporns, 2012; Sporns & Betzel, 2016), BACE estimated stronger ipsilateral than contralateral connections across all contexts. On average, the mean magnitude of ipsilateral effective connectivity was 9–16% greater than that of contralateral connections (Fig. 4C), with statistical testing reported in Appendix C. Second, we assessed hemispheric asymmetry expected from contralateral motor organization: because subject performed the task with the left hand, greater outgoing influence from the right hemisphere was anticipated during movement (Shibasaki & Hallett, 2006; Vingerhoets, 2014). In result, hemisphere-of-origin analysis showed right side-dominant connectivity in React, Reach, and Return, whereas Wait remained balanced (Fig. 4B). Together, these spatial patterns highlight that BACE not only achieves strong predictive accuracy but also produces interpretable connectivity estimates that align with some established organizational principles of brain networks.

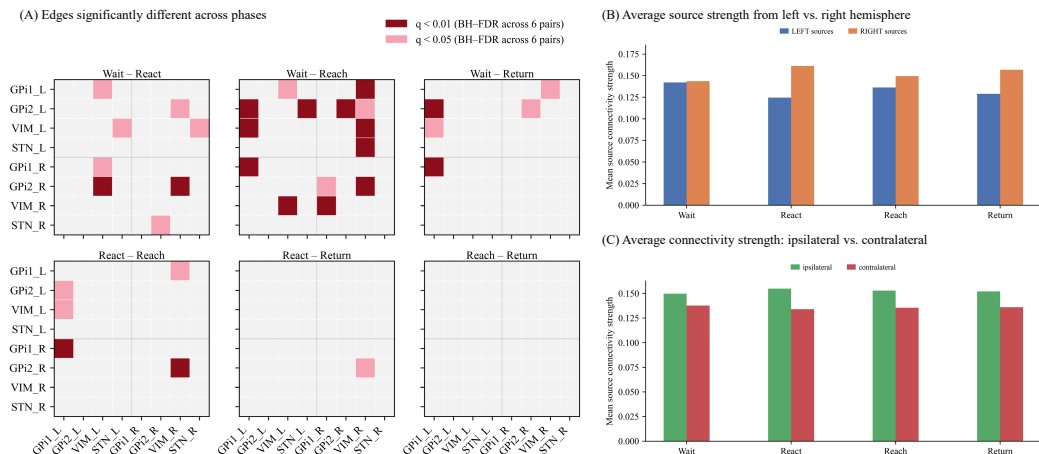

Figure 4: **Group-level analyses of behaviorally-specific connectivity.** (**A**) Edges showing significant differences across behavioral conditions, based on bootstrap CIs and Benjamini–Hochberg false discovery rate (FDR) correction. (**B**) Average outgoing connectivity strength from left vs. right hemisphere sources. (**C**) Average ipsilateral vs. contralateral connectivity strength.

## 5 DISCUSSION

This work introduced BACE, a framework for estimating behavior-adapted connectivity from multi-region neural recordings. BACE couples forecasting with graph estimation, learning compact, directed inter-regional graphs whose parameters encode predictive interactions. By operating at the region level, the model consolidates many micro-contacts into interpretable embeddings and then estimates connectivity among regions. The resulting adjacency matrices are behavior-specific, and interpretable, supporting network analyses that can be inspected by domain experts.

In synthetic experiments, BACE recovered ground-truth connectivity through forecasting, indicating that dynamics-driven objectives can surface interaction structure in multivariate time series. Applied to human subcortical LFPs during a cued reaching task, the model produced one connectivity matrix per behavioral segment. Comparisons revealed segment-dependent reconfiguration of inter-regional influence, suggesting that deep-brain networks adapt their directed connectivity in a behavior-selective manner. The organization also reflected established principles: stronger ipsilateral than contralateral connections, consistent with wiring-cost and modularity constraints (Bullmore & Sporns, 2012; Sporns & Betzel, 2016), and right-hemisphere dominance during left-hand movement, in line with contralateral motor organization (Shibasaki & Hallett, 2006; Vingerhoets, 2014). Methodologically, several design choices contribute to interpretability. Independent regional encoders prevent cross-region leakage prior to graph mixing, adjacency parameterization separates relative edge patterns from broadcast strength, and lightweight temporal regularizers stabilize short-horizon predictions.

While this study provides a proof of concept, it remains limited to a single participant and should be viewed as a case study rather than a population-level analysis. The formulation also assumes labeled behavioral segments, which constrains applicability to datasets without clear trial structure, and the estimated graphs capture effective, predictive influence rather than causal interactions. Nonetheless, these choices allow us to focus on a well-controlled setting and demonstrate feasibility. Future directions, generalization across participants and tasks, integration of anatomical priors (e.g., diffusion tractography) or multimodal signals (EMG, video), and extensions toward continuously time-varying or weakly supervised contexts would broaden applicability. Incorporating causal discovery objectives could further enhance interpretability, and the framework is not specific to LFP: the same region-level design could be adapted to EEG, ECoG, or other neural modalities. Clinically, if replicated across patients and disorders, behavior-adapted connectivity estimates may offer a foundation for hypothesis generation and, in the longer term, inform exploratory simulation studies of deep-brain stimulation strategies, though any translational use will require substantial validation.

## ETHICS STATEMENT

Human neural data were collected under IRB-approved protocols at participating medical centers. Informed consent was obtained from participants or legal guardians for minors. All data were de-identified before analysis, stored on secure access-controlled servers, and used solely for research in compliance with HIPAA and institutional privacy policies. The study did not involve clinical decision-making, non-standard interventions, or recruitment beyond patients already evaluated for DBS, and no re-identification was attempted. Raw recordings cannot be publicly shared due to ethical and regulatory restrictions, but de-identified datasets may be provided upon reasonable request under IRB-approved data use agreements. The authors confirm adherence to the ICLR Code of Ethics and declare no conflicts of interest or external sponsorship that could have influenced the research.

## REPRODUCIBILITY STATEMENT

Code for model implementation, training, and evaluation is available at `https://github.com/iclr26anon/BACE.git`.

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

# A    METHOD DETAILS

In the main text we denoted the context-specific adjacency by $\mathbf{A}^{(k)}$, a directed $N \times N$ matrix. Concretely, we parameterize it as

$$\mathbf{A}^{(k)} = \mathrm{Diag}(\mathbf{g}^{(k)})\,\bar{\mathbf{S}}^{(k)},$$

where $\bar{\mathbf{S}}^{(k)}$ is a signed pattern matrix with zero diagonal and rows normalized to unit $\ell_1$ norm ($\sum_j |\bar{S}_{ij}^{(k)}| = 1$), and $\mathbf{g}^{(k)} \in \mathbb{R}_{\geq 0}^N$ is a vector of non-negative row gains. Row normalization of $\bar{\mathbf{S}}^{(k)}$ ensures stability and makes each row interpretable as a distribution of relative influences onto region $i$, while the row gains $\mathbf{g}^{(k)}$ restore the ability of different regions to vary in their overall incoming strength. Sparsity is encouraged through an $\ell_1$ penalty on the off-diagonal entries of $\bar{\mathbf{S}}^{(k)}$.

## A.1    FORECASTING HEAD: IMPLEMENTATION DETAILS

**Attention context.**    At each forecast step $t$, the decoder forms a query from its hidden state $\mathbf{h}_{t-1}$ and computes attention weights over the encoder sequence $\mathbf{Z}$ of length $T_{\text{in}}$:

$$\mathbf{c}_t = \sum_{\tau=1}^{T_{\text{in}}} \alpha_{t,\tau}\, \mathbf{Z}_\tau, \quad \alpha_{t,\tau} = \frac{\exp(\mathbf{q}_t^\top \mathbf{k}_\tau)}{\sum_{\tau'} \exp(\mathbf{q}_t^\top \mathbf{k}_{\tau'})}.$$

**Adjacency feedback.**    The previous prediction is refined by adjacency mixing:

$$\tilde{\mathbf{y}}_{t-1,i} = \hat{\mathbf{y}}_{t-1,i} + \alpha \sum_j A_{ij}^{(k)}\, \hat{\mathbf{y}}_{t-1,j},$$

where $\alpha$ is a small damping coefficient. This step enforces that autoregressive forecasts remain connectivity-aware.

**GRU update.**    The decoder state is updated as

$$\mathbf{h}_t = \mathrm{GRUCell}\big([\mathbf{c}_t, \tilde{\mathbf{y}}_{t-1}], \mathbf{h}_{t-1}\big),$$

where $[\cdot, \cdot]$ denotes concatenation.

**Residual decoding.**    The output increment is computed as

$$\Delta \hat{\mathbf{y}}_t = W_o \mathbf{h}_t,$$

and absolute forecasts are recovered recursively:

$$\hat{\mathbf{y}}_t = \hat{\mathbf{y}}_{t-1} + \Delta \hat{\mathbf{y}}_t, \quad \hat{\mathbf{y}}_0 = \mathbf{x}_{\text{last}}.$$

**Final readout.**    Regional predictions are mapped back to the channel level by a linear projection, yielding

$$\hat{\mathbf{Y}} \in \mathbb{R}^{N \times C \times T_{\text{out}}}.$$

## A.2    TRAINING OBJECTIVE AND OPTIMIZATION

**Forecasting objective.**    Given past window $\mathbf{X} \in \mathbb{R}^{N \times C \times T_{\text{in}}}$ and future target $\mathbf{Y} \in \mathbb{R}^{N \times C \times T_{\text{out}}}$, the decoder outputs increments $\Delta \hat{\mathbf{Y}}$ which are accumulated to produce forecasts

$$\hat{\mathbf{Y}}_t = \mathbf{X}_{\text{last}} + \sum_{\tau=1}^t \Delta \hat{\mathbf{Y}}_\tau,$$

where $\mathbf{X}_{\text{last}}$ is the final observed input sample. This residual formulation mitigates drift in multi-step prediction.

**Loss function.**    The full training loss is

$$
\mathcal{L} = \underbrace{\frac{1}{T_{\text{out}}} \sum_{t=1}^{T_{\text{out}}} w_t \big\| \hat{\mathbf{Y}}_t - \mathbf{Y}_t \big\|_2^2}_{\text{forecasting MSE}} + \lambda_S \,\big\| \bar{\mathbf{S}}^{(k)} \big\|_{1,\text{off}}
$$

$$
+ \lambda_{\text{cont}} \,\big\| \hat{\mathbf{Y}}_1 - \mathbf{X}_{\text{last}} \big\|_2^2 + \lambda_{\text{vel}} \,\big\| \Delta \hat{\mathbf{Y}}_1 - \Delta \mathbf{Y}_1 \big\|_2^2 + \lambda_{\text{curv}} \,\big\| \Delta^2 \hat{\mathbf{Y}} - \Delta^2 \mathbf{Y} \big\|_2^2.
$$

$$(1)$$

- The first term is the mean squared error across the horizon, with log-spaced weights $w_t$ emphasizing later steps.
- $\| \bar{\mathbf{S}}^{(k)} \|_{1,\text{off}}$ applies an $\ell_1$ penalty on the off-diagonal entries of the normalized pattern matrix, promoting sparse graphs while leaving row gains $\mathbf{g}^{(k)}$ unconstrained.
- Continuity, velocity, and curvature penalties regularize short-term rollouts by aligning the first prediction with the last input, matching the initial slope, and constraining higher-order differences.

**Optimization.**    We use Adam (learning rate $10^{-3}$, weight decay $10^{-4}$), gradient clipping at $1.0$, and early stopping with patience 10. Graph parameters are initialized from train-only regionwise correlations (diagonal zeroed), and row gains initialized near one to preserve scale.

# B    ADDITIONAL DETAILS ON EXPERIMENTS

## B.1    APPENDIX: DATASET AND PREPROCESSING

**Participant and Experimental Protocol.**    Data were recorded from an 11-year-old male with dyskinetic cerebral palsy approximately five days after temporary deep brain stimulation (DBS) lead implantation surgery. The diagnosis of dystonia was confirmed by a pediatric movement disorder specialist using standard criteria (Anonymized, Anonymized). The study was approved by the local Institutional Review Boards, with informed consent obtained from the patient's legal guardians.

During the experiment, the subject performed a cued reaching task. At the beginning of each trial, the subject held a "home" button. After a random interval (500–1500 ms), a green "go" cue was presented, prompting the subject to release the home button and reach forward to press a target button. After pressing the target, the subject was instructed to release it and return the hand to the home button to prepare for the next trial. Of 350 attempted trials, 326 valid trials were retained for analysis after exclusion based on task compliance (Fig. 3).

**Recording Electrodes and Acquisition.**    Recordings were obtained from up to eight temporary stereoelectroencephalography (sEEG) depth leads (AdTech MM16C), implanted into candidate DBS targets using standard stereotactic procedures (Anonymized, Anonymized). Each lead included six low-impedance macro-contacts (not used in this study) and ten high-impedance micro-contacts ($50\,\mu$m, 70–90 k$\Omega$). Signals were digitized at 24,414 Hz with a PZ5M 256-channel digitizer and RZ2 processor, and stored on an RS4 system (Tucker-Davis Technologies) (Anonymized, Anonymized). Only local field potentials (LFP) from the micro-contacts were analyzed.

**Preprocessing.**    All signals were low-pass filtered at 50 Hz, re-referenced to the common average, and downsampled to 1 kHz to yield LFP time series suitable for analysis (Anonymized, Anonymized).

**Trial Segmentation.**    Each trial was segmented into four behavioral segments of equal length (400 ms):

- **Wait:** 400 ms preceding the go cue onset.
- **React:** 0–400 ms following cue onset, prior to movement initiation.
- **Reach:** from reach onset to 400 ms after.
- **Return:** from return onset to 400 ms after.

This segmentation ensured consistent windows across trials aligned to behavioral events.

**Sliding Windowing and Data Splitting.** Within each behavioral segment, data were divided into overlapping windows of length $T_{in} = 100$ samples for input and $T_{out} = 20$ samples for forecasting, with stride 20. The dataset was split into training, validation, and test sets at the *trial* level (not at the window level) to prevent data leakage. Normalization parameters (mean and standard deviation) were computed on the training set and applied consistently across validation and test sets.

## B.2 ADDITIONAL MODEL DETAILS

**Details of Comparison Models.**

**LSTM.** We build a two-layer LSTM forecaster with hidden dimension 128. The model takes past windows of length 100 across all 80 channels and predicts the next 20 steps jointly for every channel. Training is performed with mean squared error loss using the Adam optimizer, with learning rate $10^{-3}$, weight decay $10^{-4}$, and gradient clipping at 1.0. Models are trained for up to 100 epochs with early stopping (patience of 10, minimum delta $10^{-4}$), and the best checkpoint is selected based on validation loss. Evaluation reports mean squared error on the test set.

**NDT.** We implement a Neural Data Transformer (NDT) baseline under a masked-forecasting setup. While NDT is typically used to reconstruct randomly masked neural activity, we adapt it for forecasting by designating the final 20 steps of each window as the masked segment. The model is a three-layer Transformer with attention dimension 128, applied over windows of length $100+20$ across all channels. During training, the past 100 bins are provided while the last 20 bins are zero-masked, and the model is optimized to reconstruct only this masked future segment. Training uses the Adam optimizer with learning rate $10^{-4}$, weight decay $10^{-5}$, and early stopping based on validation loss.

**Graph WaveNet.** We implement Graph WaveNet (GWNet) to benchmark forecasting performance on the Button Task LFP dataset. The architecture consists of three layers, each composed of two spatio-temporal blocks (six blocks in total). Each block integrates (i) a gated dilated temporal convolution, (ii) a graph convolutional module for spatial mixing, and (iii) residual and skip connections. All temporal convolutions use kernel size 2, with 64 channels in the residual and dilated paths and 128 channels in the skip path. The skip outputs are aggregated and passed through a two-stage output head consisting of a $1 \times 1$ convolution with 64 channels followed by a read-out convolution mapping to the 20-step forecasting horizon. Training is performed using the Adam optimizer with an initial learning rate of $5 \times 10^{-4}$, decayed by 0.95 every 50 epochs, and no weight decay. Static graph supports are constructed from training data via correlation-based $k$-nearest neighbors, and an adaptive adjacency matrix is learned jointly during training.

**DCRNN.** We implement the Diffusion Convolutional Recurrent Neural Network (DCRNN) using a sequence-to-sequence architecture with diffusion convolutional GRU (DCGRU) cells. Both encoder and decoder consist of two stacked DCGRU layers with hidden dimension 64. Each DCGRU layer performs diffusion convolution with $K=2$ steps. The model is trained with the Adam optimizer (learning rate $5 \times 10^{-4}$, decayed by 0.95 every 50 epochs), no weight decay, and scheduled sampling (inverse-sigmoid with $\tau=3000$).

**AMAG.** For multi-step forecasting, we implement AMAG with a single-layer Transformer encoder and decoder, each with hidden size 64 and two attention heads. Both encoder and decoder operate on 80-channel inputs with positional encodings. The sample-dependent adaptor MLP takes concatenated hidden features from node pairs and applies four sequential fully connected layers of sizes $64 \times t$, $64 \times 2$, $64 \times 4$, and 64, followed by a scalar output with sigmoid activation. Training uses the Adam optimizer with an initial learning rate of $5 \times 10^{-4}$, decaying by 0.95 every 50 epochs, with early stopping based on validation loss.

## B.3 COMPUTE RESOURCES

All models were trained on a single workstation equipped with an AMD Ryzen Threadripper PRO 7955WX CPU (16 cores), 128 GiB RAM, and one NVIDIA RTX 4500 Ada GPU (24 GiB VRAM). Training was performed using PyTorch 2.2 with CUDA 12.2.

Table 2 in the main text summarizes the parameter counts and peak memory usage for each model. Average epoch training times are reported alongside in the results section.

## C  STATISTICAL INFERENCE OF CONNECTIVITY

While forecasting accuracy serves as the primary training signal, we further evaluate the stability and significance of the learned graphs.

**Split–half reproducibility.** For each behavioral context $k$, we performed $K=50$ random half–splits of trials. Graph parameters $(\mathbf{S}^{(k)}, \mathbf{g}^{(k)})$ were re–estimated on each half, and Pearson correlations of $|\mathbf{A}^{(k)}|$ were computed. Spearman–Brown correction extrapolated full–data repeatability. Mean split–half correlations (95% confidence intervals) were: Wait $r=0.90$ [0.83, 0.94], React $r=0.88$ [0.82, 0.93], Reach $r=0.91$ [0.86, 0.94], Return $r=0.83$ [0.75, 0.89]. The corresponding Spearman–Brown coefficients were 0.95, 0.94, 0.95, and 0.91, respectively. These values indicate that context-specific graphs are highly reproducible across independent subsets.

**Bootstrap resampling.** For precision of individual edges, we performed $B=1000$ nonparametric bootstrap resamples per context, refitting only graph parameters while holding encoder/decoder weights fixed. For each edge $(i, j)$, we computed percentile 95% confidence intervals of $|A_{ij}^{(k)}|$. An edge was considered reliable if its CI excluded zero and its sign was consistent in at least 80% of resamples.

**Edgewise precision.** Absolute CI widths were narrow and consistent across contexts (medians $\approx 0.05$–$0.07$; few edges $> 0.10$). To provide a scale–free summary, we report the relative half–width

$$\text{rHW} = \frac{\text{hi} - \text{lo}}{2\,|\text{median}|},$$

evaluated on edges whose median magnitude exceeded a threshold $\tau$ (66th percentile). Median rHW values were $\approx 0.12$–$0.20$, meaning nontrivial edges had 95% intervals of about $\pm 12$–$20\%$ of their estimate.

**Across–context differences.** To detect reconfiguration, we computed bootstrap distributions of edgewise differences

$$\Delta_{ij} = |A_{ij}^{(k_1)}| - |A_{ij}^{(k_2)}|,$$

applied two–sided sign tests, and corrected using Benjamini–Hochberg FDR ($\alpha=0.05$). Fig. 4A displays only those edges whose CIs excluded zero and survived correction. Complete edgewise statistics (median $\Delta$, CI, $p$, $q$) are reported in the supplement.

**Ipsilateral vs. contralateral.** We quantified lateral organization by comparing average magnitudes of ipsilateral and contralateral connections. A one–sided paired bootstrap test ($B=1000$) on the log–ratio confirmed stronger ipsilateral connectivity in three of four contexts (Wait $p<0.05$, React $p<0.001$, Reach $p<0.01$).

**Hemisphere dominance.** For lateralized motor organization, we averaged outgoing effective connectivity $\overline{|A_{\text{eff}}^{(k)}|}$ from each hemisphere. Right–hemisphere dominance emerged during movement contexts (React, Reach, Return) but not Wait. A complementary bootstrap test on the log–ratio is reported in the supplement.

**The Use of Large Language Models (LLMs).** LLMs were used as a general-purpose tool for editing, code debugging, and literature organization. Their role was limited to improving clarity of writing, drafting boilerplate sections, and suggesting implementation fixes. All research questions, methods, analyses, and conclusions were developed by the authors, and the LLM is not considered a scientific contributor.

