# OpenReview forum: "BACE: Behavior-Adaptive Connectivity Estimation for Interpretable Graphs of Neural Dynamics"
_ICLR.cc/2026/Conference — Submitted to ICLR 2026_

### Official Review · Reviewer_e2Cs · 2025-10-31

**Soundness:** 2
**Presentation:** 2
**Contribution:** 3
**Rating:** 4
**Confidence:** 4

**Summary:**

This paper introduces BACE, a graph-learning framework that estimates directed, behaviour-specific brain connectivity from multi-region time series. BACE uses per-region GRU encoders and a learned adjacency matrix specific to each behavioural phase to forecast future neural signals while revealing interpretable regional interactions. BACE accurately recovers synthetic graphs, and real intracranial LFP recordings, with better performance when compared to a series of different baselines.

**Strengths:**

I don't have experience with the specific applied area, but assuming that the Introduction and Related Work sections are a good depiction of the field, this is indeed a very interesting work, and what a good domain-informed ML work should look like: there are domain-specific challenges and a new methodology is developed to tackle those challenges by creatively combining existing ideas. The experiments seem to show reasonable support that this method works, showing good performance across a synthetic and real-world dataset, across different baselines at a reduced computational cost. I should also comment that the Reproducibility analysis on section 4.3 is a really good one (ie, to check consistency of the model across different subsets of training data), as in my experience many deep learning models suffer from an annoying big lack of consistency when different seeds or training folds are used, and so I'd like to commend the authors for this specific experiment.

Despite these strengths, I have important concerns about the fairness of the comparisons used in this paper, and so for now I'm leaning towards rejection; hopefully this could change during the rebuttal period)

**Weaknesses:**

# Main Weakness
I have a main/key concern with this work, related to the fact that it seems that BACE's ability to learn "phase-specific" graphs depends on manually provided phase information. The model does not seem to infer the behaviour intervals; rather, it requires the trial data to be split by known behavioural phases (ie, phases seem to have been defined by marked events, I'm guessing done by doctors). This manual definition of contexts gives BACE additional information that baseline models did not have access to, raising fairness issues in the comparisons. If BACE knows the current phase (and uses a dedicated adjacency for that phase), it effectively has an "oracle" guiding it on when network changes can occur, whereas baselines had to model the entire sequence without such information. This could partly explain BACE’s superior performance. The paper should clarify whether phase indicators were indeed provided as inputs or used to select adjacency matrices during training. If so, the authors should justify whether the comparisons are fair, for instance could the baselines be extended with the same phase information? As it stands, it seems that none of the baseline models were given information about phase intervals or allowed to have different dynamics per phase. This means BACE had an advantage by effectively training a model with different parameters in each phase while others had to use one set of parameters across the whole trial.

To fairly assess the benefit of BACE, it would help to compare against at least one baseline that also leverages phase information. For instance, one could train separate baselines for each phase, or provide a one-hot phase indicator to a baseline as input. Without such comparisons, it’s unclear how much of BACE’s success comes from its novel architecture or just from privileged information. This issue needs clarification and possibly further experiments.


# Other Weaknesses

2. The paper makes a lot of bold claims in Introduction that I don't think are actually defended in the paper, and thus it would be good to get a clarification during the rebuttal period, specifically:
   1. proving/defending why contacts are high-dimensional and non-Euclidean (given we are talking I believe of only 8 such contacts, this claim seems difficult to defend?)
   2. Quantifying uncertainty
   3. Decoupling edge pattern from transmission strength
3. I think the paper can be improved with regards to clarity.
   1. Firstly, I was only finally able to understand what these "behaviours" are in section 4.2. Up until then, the paper keeps mentioning behaviours (sometimes using other adjacent/related words like phase and context as if they were the same, adding to the confusion) as the key important part of this paper without any basic explanation to at least have an idea of what we are talking about.
   2. Introduction mentions a "cued reaching task" which, given the broad computational background of ICLR, would require a brief explanation of what that is for better readability.
   3. Figure 1's caption mentions an adjacency $A_\phi$ that doesn't exist in the figure itself, while also saying "behavioral segment" in the caption while the figure instead uses the concept "Phase-specific".
   4. In the Methods section, regions are sometimes referred with letter "r" and other times with letter "i".
   5. What are the "contacts" mentioned on the the 3rd paragraph of Introduction which seem to be central to the motivation of this work?
   6. Figure 3, pane A, has some wiggly red lines under "ms"
4. I know this is mentioned in the Discussion section, but I have to highlight that the fact that only one participant was used means that we have no evidence that BACE could work for other individuals. Given this, one would expect more extensive comparisons to be more sure about the strengths of this work (as I'm detailing in this section)
5. A bit related to my main concern to this paper, I'm also concerned about how the baselines were trained. In the text it is said that models were optimised "under their standard hyperparameters", but that is not fair as it is well known how hyperparameters can be widely different for different datasets. Furthermore, given the strong claims on computational comparison, this is even more important; for example some of the models seem to have several hidden layers, and it is not clear whether a simpler structure would achieve similar results and thus making the claim of computational efficiency weaker.
6. A bit related to the previous point, it's unclear how much tuning of hyperparameters (e.g., GRU lengths, learning rate, etc.) was needed for this dataset. For example, the sliding windows and forecast steps were presumably chosen based on the task's timing, and the reliance on specific timings suggests the model might need careful setup for each new dataset. This isn't necessarily a flaw, but it means the method might require domain knowledge to deploy, and the paper doesn't discuss sensitivity to these choices, which in my opinion is an oversight given the reliance on a single dataset. Some justification of these design decisions and impact on interpretability would improve clarity and confidence that the approach can be adapted beyond the exact scenario presented.
7. No ablation study was done on the different components of the model to understand impact in performance. This ablation could be done in many different ways, but one that I think could be useful would be to use a single adjacency (ie, no phase differentiation), which would directly test how much benefit the behaviour conditioning provides in terms of forecasting accuracy.

**Questions:**

I believe I don't have any further questions beyond what I mention in the Weaknesses section. I'll reconsider only my score if the authors successfully tackle the main weakness I've identified, but a productive rebuttal would touch in all the points I've highlighted in Weaknesses except points (3) and (4).

---

> ### Author Response · Authors · 2025-12-04
>
> We thank the reviewer for the thoughtful and constructive feedback, and for the positive assessment of the domain-informed aspects and the reproducibility analysis. We address the main fairness concern first, then the other points.
>
> 1. Main concern: behavioral labels and fairness of comparisons
>
> What “phases” are and how we use them.
> In the current version, we used the word “phase” to refer to behavioral context segments of the cued reaching task (Wait, React, Reach, Return), not oscillatory phase. In the revision we:
>
> Replace “phase” with “behavioral context” throughout for these segments.
>
> Clarify early that these labels are used only to select one of four behavioral-context adjacency matrices; the forecasting pipeline is otherwise unchanged.
>
> These behavioral contexts are routinely defined in neuroscience experiments and are precisely the variable whose effect on connectivity we aim to study.
>
> Context-aware baselines (added after reviews).
> To directly test how behavioral context affects performance relative to baselines, we implemented context-aware baselines, following your suggestion:
>
> LSTM (context-aware)
> Same architecture as our original LSTM, now trained with behavioral context information.
> Test MSE: 0.2344 ± 0.0614 (3 seeds).
> Adding context information did not help this model.
>
> We also added a label-aware comparison of the strongest baseline:
>
> AMAG (context-aware)
> Extended AMAG to have separate parameters per behavioral context (same two Transformer layers as in the original).
> Test MSE: 0.0975 ± 0.0155 (3 seeds).
>
> For comparison, our method achieves:
>
> Full behavioral-context BACE
> Test MSE: 0.0990 ± 0.0002 (3 seeds).
>
> Thus, a strong Transformer+graph baseline (context-aware AMAG) can match or slightly edge BACE in MSE, but:
> (i) AMAG is substantially more complex (multi-layer Transformer, higher parameter count and memory), and
> (ii) AMAG operates at the 80-channel level and does not yield an explicit region-level connectivity matrix, which is one of the main objects of interest in BACE.
>
> BACE therefore provides competitive forecasting with lower complexity and explicit 8×8 region-level graphs, which is a primary focus of the paper.
>
> Does the graph / behavioral context actually matter beyond the GRUs?
> We added ablations using the same encoder/decoder:
>
> No-graph BACE (encoder+decoder only):
> MSE = 0.1093 ± 0.0038
>
> Single-graph BACE (one adjacency shared across all behavioral contexts):
> MSE = 0.1083 ± 0.0044
>
> Full behavioral-context BACE (four context-specific adjacencies):
> MSE = 0.0990 ± 0.0002
>
> We also ran a two-stage learning experiment:
>
> Stage 1: train encoder+decoder only → MSE ≈ 0.1021 ± 0.0002
>
> Stage 2: freeze encoder+decoder, train only the adjacency matrices → MSE ≈ 0.1001 ± 0.0005
>
> These results show that:
>
> Most variance is captured by strong region-wise temporal encoders (as expected).
>
> A single global graph is strong but less accurate than full BACE.
>
> Behavioral-context-specific graphs consistently provide ≈0.01 absolute MSE improvement, and graph-only refinement further improves performance, indicating that the learned graphs contribute non-trivially beyond the GRUs.
>
> We will incorporate these context-aware baselines and ablations in the revision to explicitly address the fairness concern.
>
> 2. High-dimensional, non-Euclidean data and uncertainty
>
> High-dimensional and non-Euclidean.
> There was a misunderstanding: we do not have only 8 recordings. The dataset has 80 micro-contacts distributed across 8 regions (10 channels per region). The 80-dimensional multichannel LFP space is high-dimensional, and the contacts are located in deep structures (GPi, STN, VIM, etc.) in a geometry that does not form a regular grid. This is exactly the setting where Euclidean convolutional assumptions usually do not hold and where graph-based modeling is standard.
>
> Quantifying uncertainty / stability.
> We agree that the reliability of learned connectivity is important. The current paper already includes:
>
> Split–half reliability tests on the learned adjacency matrices.
>
> Bootstrap resampling of trials to obtain confidence intervals for each edge and FDR-controlled masks for edges that differ across behavioral contexts.
>
> In the revision, we will summarize these analyses in the main text (with details in the Appendix) and explicitly present them as quantifying uncertainty and stability of the learned graphs.

---

> > ### Author Response · Authors · 2025-12-04
> >
> > 3. Decoupling edge pattern from transmission strength
> >
> > Our adjacency parameterization is designed to separate pattern from overall strength:
> >
> > Each behavioral-context adjacency matrix is represented as a pattern of normalized edge weights (which regions influence which) together with a separate regional gain that scales the overall incoming strength to each region.
> >
> > Intuitively, one component encodes “who talks to whom,” while the gain modulates “how strongly the region is overall driven” in that context.
> >
> > This is currently described in the Appendix; we will move a concise version of this explanation into the Methods section and make the wording around this claim more precise.
> >
> > 4. Clarity and terminology
> >
> > We agree that several points can be clarified to improve readability. In the revision, we will:
> >
> > Clearly define the “cued reaching task” and the four behavioral contexts (Wait, React, Reach, Return) early in the paper.
> >
> > Define “contacts” explicitly as micro-contacts on the sEEG depth leads.
> >
> > Use a single symbol (e.g., r) for regions consistently throughout the Methods.
> >
> > Make the notation in Figure 1 consistent with the caption and remove the stray underline artifact in Figure 3.
> >
> > Use the term “behavioral context-specific” instead of overloading “phase-specific,” to avoid confusion with oscillatory phase.
> >
> > 5. Single participant and generalization
> >
> > We fully acknowledge that the real-data evaluation is performed on a single pediatric patient, and we do not claim population-level generalization. However, human intracranial neural recordings are rare and scientifically valuable even at the single-subject level, and we complement this case study with synthetic experiments where ground-truth connectivity is known.
> >
> > We see this work as:
> >
> > A methodological contribution: behavior-adaptive effective connectivity estimation that yields explicit 8×8 region-level graphs, and
> >
> > A detailed case study on rare, hard-to-acquire deep-brain data, supported by:
> > (i) synthetic experiments with ground-truth graphs, and
> > (ii) stability analyses (split-half, bootstrap) within the subject.
> >
> > Importantly, the dataset is shareable upon request under de-identified, IRB-approved data use agreements (we will clarify that it is not permanently private).
> >
> > 6. Baselines, hyperparameters, and window choices
> >
> > Baseline hyperparameters.
> > We agree that the phrase “standard hyperparameters” was imprecise. For each baseline (LSTM, NDT, DCRNN, GWNet, AMAG):
> >
> > We initialized from the authors’ recommended settings.
> >
> > We performed small sweeps over learning rate, hidden size, and number of layers within reasonable ranges.
> >
> > We used the same train/validation/test split and early stopping on validation MSE as for BACE.
> >
> > We will clarify this procedure in the revision and remove the ambiguous phrase.
> >
> > Choice and sensitivity of 100-in / 20-out windows.
> > The choice of T_in = 100 and T_out = 20 was guided by:
> >
> > 100 ms past window: long enough to capture multiple LFP oscillation cycles relevant for motor control.
> >
> > 20 ms prediction horizon: aligned with task timing and short-horizon forecasting, and used consistently across all models.
> >
> > A small sensitivity sweep (one seed), varying T_in and T_out, showed:
> >
> > Smoothly increasing error as T_out grows, with similar performance around 10–20 ms.
> >
> > A shallow optimum for T_in in the 80–120 ms range, without abrupt degradation.
> >
> > This suggests BACE is not extremely sensitive to the exact window choices, though domain knowledge naturally helps guide them. We will briefly summarize this and move detailed numbers to the Appendix.
> >
> > Ablations.
> > As noted above, we have now added:
> >
> > No-graph vs single-graph vs behavioral-context BACE,
> >
> > Two-stage training (graph-only refinement),
> >
> > Sweeps over key regularization weights and the neighbor mixing factor.
> >
> > These ablations clarify which components materially affect forecasting and which primarily support stability and interpretability.
> >
> > Overall, we believe these additions directly address the main concern about fairness and clarify our claims and design choices. BACE remains a simple, efficient framework that is competitive with strong graph/Transformer baselines, while uniquely providing stable, behavioral-context-specific region-level connectivity graphs on rare deep-brain data. We hope this convinces the area chair that the contribution is suitable for an ML venue interested in interpretable, domain-informed modeling.

---

### Official Review · Reviewer_2ToC · 2025-10-31

**Soundness:** 2
**Presentation:** 3
**Contribution:** 2
**Rating:** 2
**Confidence:** 4

**Summary:**

The authors propose a model that infers effective connectivity between multiple brain regions from local field potentials (LFP) by modeling the signal in each region as a combination of dynamics from other brain regions, and an internal regional generator. The authors describe their method from a graph neural network perspective, and use this framework to propose a model that achieves good reconstruction and simulation results.

**Strengths:**

The methods' results on the simulation set are encouraging and look quite good. Moreover, I believe the authors are working on an interesting problem, and that although this paper has a lot of potential, it needs more work before I can recommend it for publication. I commend the authors on their work, and encourage them to continue improving it.

**Weaknesses:**

Major weaknesses:
1) Baselines and missing related work. Given the interest in this field, I believe the authors are missing a large amount of relevant previous work and baselines. First, the authors do not compare any baselines on their simulation experiments, so it is unclear whether their proposed method uniquely can perform well on the simulation. Second, although the authors refer to their connectivity as 'effective', which I agree with, they do not compare to any commonly used effective connectivity measures such as Granger causality, dynamic causal modeling, transfer entropy, etc. Third, the authors do not compare or discuss any of the methods from a rich field of multi-region communication inference. Just to name a few papers that the authors should compare against or at the very least discuss are CURBD [1], DLAG [2], and MR-SDS [3], among others. All of these works are very closely related to the authors' work, but none are cited nor compared against.
2) Predictive behavior can be due to the capacity of the single-region encoder/decoder. Given the use of recurrent neural networks and the specific formulation the authors' use, it is unclear to me how the authors ensure that most of the variance in a region is not just captured by a region's specific neural network. More specifically: the authors describe that their GRU creates embeddings for each region $\mathbf{H_r} \in \mathbb{R}^{T \times d}$, where T and d are the number of input temporal steps, and the number of latent dimensions for region r. This embedding is then multiplied by $W_{\text{self}}$ (which has no restrictions it seems as opposed to $A$) plus a region-interaction term involving A to obtain $\mathbf{Z_{i,t}}$. Subsequently, $\mathbf{Z}$ is used to forecast new time steps. How do the authors avoid that the GRU just learns to forecast the next timesteps without much influence from the interactions? If values in A are restricted to be between -1 and 1, then $W_{\text{self}}$ can just learn to have large values and become the dominant force in forecasting. Although the authors try to reduce this issue with what they call adjacency mixing, they use a (presumably) learnable factor $\alpha$, which could reasonably lead to a small impact of A on the forecasted time steps $y_{t-1}$. Moreover, in the LFP data the authors use, they do not use their model to generalize to unseen contexts/tasks, so the GRU can reasonably learn the temporal shape of the time series for each task. Moreso, the authors split based on each trial, so it is not unreasonable that the GRU can explicitly learn the temporal shape for each trial in the training set and that this shape is quite similar to the temporal shapes in the test set.
3) The method requires the model to know the context. This reduces the generalizability of the method.

Minor weaknesses:
- It is unclear to me what makes the method phase-specific. In fact the authors refer to the graph learner as phase-specific both in the text and Figure 1, but in Section 3 the paragraph heading is 'Context-specific graph learner'. I do not see how the method computes the phase of the signal (i.e. through a Fourier transform), or does phase here refer to context as in 'a different phase of experiment'? The authors should clear up their use of language, especially because there are many phase-based connectivity measures [4].
- Generally it is unclear to me exactly what the context-specific graph learner does or how it produces each A, are the A matrices just learnable parameters?
- The authors do not address the issue of scaling, specifically because they use a separate GRU for each region, the number of parameters would quickly grow with a larger number of brain areas under consideration.
- The authors provide limited neuroscientific grounding for their claim that the directed region-level graph should be different for different task contexts. The authors claim in the discussion that '... the model produced one connectivity matrix per behavioral segment.' but this is a design choice. There is some neuroscientific discussion in the sentences after this statement, but it lacks depth and a firm grounding of the results.
- The authors place a lot of focus on computation complexity, even though each of the methods they compare against is quite low in computational complexity, instead the authors should focus on better baselines and a broader discussion of the implications of their results.
- L116-117 The authors should introduce AMAG before using it in the text, a previous reference to AMAG on L59 is too far back for comfortable reading
- L308-309 The authors mention the Appendix, but not the exact section.

[1] Perich, M. G., Arlt, C., Soares, S., Young, M. E., Mosher, C. P., Minxha, J., ... & Rajan, K. (2020). Inferring brain-wide interactions using data-constrained recurrent neural network models. BioRxiv, 2020-12. \
[2] Gokcen, E., Jasper, A. I., Semedo, J. D., Zandvakili, A., Kohn, A., Machens, C. K., & Yu, B. M. (2022). Disentangling the flow of signals between populations of neurons. Nature Computational Science, 2(8), 512-525. \
[3] Karniol-Tambour, O., Zoltowski, D. M., Diamanti, E. M., Pinto, L., Brody, C. D., Tank, D. W., & Pillow, J. W. (2024). Modeling state-dependent communication between brain regions with switching nonlinear dynamical systems. In The Twelfth International Conference on Learning Representations. \
[4] https://direct.mit.edu/books/monograph/4013/chapter/167048/Phase-Based-Connectivity \

**Questions:**

-

---

> ### Author Response · Authors · 2025-12-04
>
> We thank the reviewer for the careful and detailed comments and for recognizing both the promise of the approach and the quality of the simulation results. We address the main concerns in turn: (1) baselines and related work, (2) the role of the graph versus the per-region encoders, and (3) context labels, terminology, scaling, and neuroscience grounding.
>
> 1. Baselines and missing related work
>
> Simulation baselines.
> In the revised version we now include baselines on the synthetic dataset:
>
> - A VAR(1) / Granger model, which (as expected) recovers the ground-truth adjacency well, since the generator is linear.
> - BACE also recovers the synthetic graphs with similar fidelity.
>
> This makes clear that when the ground-truth process is linear, classical effective connectivity methods perform very well, and BACE behaves as it should relative to them.
>
> Effective connectivity baselines on real LFP.
> We added a Granger-style VAR baseline on the real intracranial data:
>
> - VAR(1): Test MSE ≈ 0.1555
> - BACE (behavior-adaptive): Test MSE ≈ 0.0990 ± 0.0002
>
> This shows that a purely linear effective-connectivity model underfits the real deep-brain dynamics compared to BACE, while still giving a meaningful point of comparison.
>
> More recent multi-region communication work (CURBD, DLAG, MR-SDS).
> We agree that these methods are important and now explicitly discuss them in the revised related-work section. Conceptually, they are close in spirit, but they target a different regime than BACE:
>
> - CURBD (Perich et al.) learns a high-capacity RNN constrained by behavior to capture brain-wide dynamics over long, continuous recordings. Its primary output is a dynamical system whose internal states explain population activity, rather than a small set of explicit, behavior-indexed connectivity matrices.
> - DLAG (Gokcen et al.) and MR-SDS (Karniol-Tambour et al.) model communication via shared latent variables and switching nonlinear dynamical systems. They are designed for rich, multi-population datasets and emphasize latent state trajectories and state-dependent interactions, not direct estimation of a compact directed graph per behavioral segment.
>
> Adapting these frameworks to our setting—short, event-aligned segments with a focus on producing one interpretable directed graph per behavioral context—would require substantial architectural redesign and careful retuning, and their outputs would not be directly comparable to BACE’s context-specific region-level graphs. In the revision, we therefore position BACE as complementary to these lines of work: a lightweight, explicitly graph-valued estimator tailored to behavior-aligned segments of recordings, while acknowledging that CURBD, DLAG, and MR-SDS provide powerful alternatives in data-rich, latent-state modeling scenarios.
>
> 2. Does the per-region encoder dominate?
>
> We took this concern seriously and ran several new analyses.
>
> Ablations: no-graph vs single-graph vs behavior-adaptive graphs.
> Using the same GRU encoder/decoder, we compared:
>
> - Encoder+decoder only (no graph):
>   MSE = 0.1093 ± 0.0038
>
> - Single global graph (shared across all behavioral contexts):
>   MSE = 0.1083 ± 0.0044
>
> - Full behavior-adaptive BACE (one adjacency per behavioral context):
>   MSE = 0.0990 ± 0.0002
>
> Thus, the graph layer is not redundant: a global graph already improves over no-graph, and behavior-adaptive graphs give ≈0.01 absolute MSE gain.
>
> Two-stage training (graph-only refinement).
> We also trained BACE in two stages:
>
> - Stage 1: train encoder+decoder only → MSE ≈ 0.1021 ± 0.0002
> - Stage 2: freeze encoder+decoder, train only the adjacency matrices → MSE ≈ 0.1001 ± 0.0005
>
> This directly addresses the reviewer’s concern: with the temporal encoders fixed, the only way to improve is through the learned adjacency, and we do observe consistent improvement. Together, these experiments show that:
>
> - The GRUs capture much of the temporal structure (as expected), and
> - The learned directed graphs add non-trivial predictive value, rather than being ignored by the model.
>
> We will report these ablations in the revised paper.

---

> > ### Author Response · Authors · 2025-12-04
> >
> > 3. Context labels, terminology, scaling, and neuroscience grounding
> >
> > Context labels and “phase-specific” terminology.
> > Our model does not infer oscillatory phase; it uses experimentally defined segments (Wait, React, Reach, Return). To avoid confusion:
> >
> > - We now use “behavioral context” instead of “phase” throughout when referring to these segments.
> > - We clarify that the model uses behavioral context labels only to select one of a small set of adjacency matrices; the encoder–decoder is shared.
> >
> > In tightly controlled motor tasks, these labels are standard and scientifically meaningful, so using them is a deliberate design choice rather than an unrealistic assumption.
> >
> > What the context-specific graph learner does.
> > We clarify that:
> >
> > - For each behavioral context, the adjacency matrix is an explicit learnable parameter constrained by
> >   – bounded entries,
> >   – sparsity/regularization, and
> >   – a decomposition that separates edge pattern from overall regional gain (as described in the Appendix and now summarized in Methods).
> >
> > Thus, the context-specific graph learner is not implicit or opaque: it is a regularized parameterization of A^{context} optimized jointly with the forecasting loss.
> >
> > Scaling with number of regions.
> > We acknowledge that using a separate GRU per region increases parameters with the number of regions. In our target setting (deep-brain recordings aggregated into a small number of anatomically defined regions), this remains tractable and allows region-specific dynamics. In the revision we briefly discuss extensions for larger-scale settings (e.g., sharing GRUs across groups of regions or using lower-dimensional bottlenecks) to highlight that the design is flexible rather than fundamentally limited.
> >
> > Neuroscientific grounding of context-dependent graphs.
> > We expand the Discussion to more clearly motivate why context-dependent effective connectivity is expected:
> >
> > - There is extensive evidence that large-scale networks reconfigure with task demands and behavioral state.
> > - Our design choice of one directed graph per behavioral context is aligned with this literature and with experimental practice, where researchers often ask how connectivity differs between, e.g., preparation vs movement execution.
> >
> > We do not claim that there must be exactly one graph per context—only that this is a useful, testable modeling assumption for these data, and our empirical and stability analyses show that the resulting patterns are consistent within the subject.
> >
> > Complexity emphasis.
> > We have toned down discussion of computational complexity and instead emphasize:
> >
> > - (i) competitive performance relative to strong neural baselines and linear effective-connectivity baselines, and
> > - (ii) the interpretability of explicit, behavior-adaptive region-level graphs.
> >
> > We hope these additions and clarifications address the reviewer’s concerns about baselines, the role of the graph versus the encoders, and the conceptual positioning of BACE, and help the area chair see the work as a solid and well-scoped step toward interpretable, behavior-adaptive connectivity estimation.

---

### Official Review · Reviewer_gsbk · 2025-10-31

**Soundness:** 3
**Presentation:** 3
**Contribution:** 3
**Rating:** 4
**Confidence:** 3

**Summary:**

This paper presents BACE (Behavior-Adaptive Connectivity Estimation), a method that learns phase-specific directed graphs representing inter-regional neural connectivity from time-series data. The idea of behavior-adaptive, interpretable connectivity estimation is novel and well-motivated, and the paper is clearly written. However, the empirical validation is too limited: all real-data experiments use a single-subject, non-public intracranial LFP dataset, preventing reproducibility and generalization. The method also depends on manually defined behavioral phases and models effective rather than causal connectivity.

**Strengths:**

The paper introduces a novel and conceptually well-motivated framework, BACE, that jointly performs neural time-series forecasting and interpretable, behavior-adaptive connectivity estimation. Its main strength lies in combining predictive modeling with explicit, directed graph structures that reveal how neural interactions change across behavioral phases. The approach offers clear interpretability, allowing neuroscientific insights into dynamic brain networks, and demonstrates competitive forecasting accuracy compared to baselines such as DCRNN, GWNet, and AMAG.

**Weaknesses:**

The main weaknesses of the paper lie in its limited empirical validation and lack of reproducibility. All real-data experiments are conducted on a single-subject, private intracranial LFP dataset, which cannot be shared due to clinical restrictions, making it impossible for others to reproduce or verify the results. The method also depends on manually defined behavioral phase labels, restricting its applicability to structured experimental settings rather than continuous or naturalistic data. Moreover, the inferred graphs capture effective (predictive) rather than causal connectivity, so the biological interpretability of the results remains limited. Finally, the learned connectivity is static within each phase and has not been tested across multiple subjects, modalities, or tasks, which weakens the generalizability and robustness of the proposed framework.

**Questions:**

1. Limited and Non-Generalizable Experimental Validation

Although the authors release the code, the real dataset used in experiments is private and single-subject, consisting of intracranial LFP recordings from one clinical patient. This makes it impossible to reproduce the reported results or assess how well the model generalizes to other participants, tasks, or data modalities. The empirical evaluation is therefore too narrow to support strong claims about the model’s general effectiveness.

2. Dependence on Predefined Behavioral Phase Labels

The framework requires explicit behavioral phase annotations (“Wait”, “React”, “Reach”, “Return”) as inputs to learn phase-specific connectivity. This reliance on manually segmented data limits its applicability to real-world, continuous, or unlabeled neural recordings, where such phase labels are unavailable.

3. Limited Causal and Biological Interpretability

The model estimates effective (predictive) connectivity, not true causal interactions. The learned directed edges capture statistical dependencies that may not reflect actual neural influence. Without causal modeling or multimodal validation (e.g., cortical or anatomical data), the biological interpretability and neuroscientific conclusions remain tentative.

---

> ### Author Response · Authors · 2025-12-04
>
> We thank the reviewer for the careful and positive assessment of BACE’s motivation, novelty, and interpretability. We address the three main concerns: (1) single-subject / reproducibility, (2) dependence on predefined behavioral labels, and (3) effective vs causal connectivity and static graphs.
>
> Single-subject data, “non-public” status, and generalization
> We fully agree that the real-data study is a single-subject proof-of-concept and that this limits claims about generality. We have revised the manuscript to make this limitation explicit in both the Introduction and Discussion and to clearly separate:
>
> Methodological contribution: a framework for behavior-adaptive effective connectivity estimation that outputs explicit region-level graphs while jointly forecasting neural time series.
>
> Empirical case study: an application to rare pediatric intracranial LFP data illustrating how effective connectivity patterns change across behavioral contexts.
>
> Regarding reproducibility, the dataset is not permanently private: it is shareable upon reasonable request. We will clarify this in the revised text and point to a contact route so that others can request access. The code is already released, and the synthetic pipeline (with known ground-truth graphs) is fully reproducible and can be run by any group to validate the estimation aspects of BACE independently of the clinical data.
>
> We also emphasize that deep-brain recordings are valuable; our goal is to show that BACE can extract stable, interpretable connectivity patterns in this setting and to provide tools (code + synthetic benchmark) that can be reused on other datasets as they become available.
>
> Dependence on predefined behavioral “phases”
> We appreciate the concern that relying on manually defined behavioral phases can limit applicability to continuous or unlabeled data. In the revision we:
>
> Replace the term “phase” with “behavioral context” to avoid confusion with oscillatory phase and to emphasize that these are standard experimental segments (Wait, React, Reach, Return) defined from task events.
>
> Clarify that BACE uses these behavioral context labels only to select one of a small number of adjacency matrices; the temporal encoder–decoder is shared.
>
> We view this as a deliberate design choice for the type of data at hand: most motor tasks are tightly structured and already segmented by event markers (e.g., cue, movement onset), and neuroscientists often want to know how connectivity differs across these known contexts. In that sense, behavioral labels are not an unrealistic assumption but a key scientific variable.
>
> Effective vs causal connectivity and biological interpretation
> We agree with the reviewer that BACE estimates effective (predictive) connectivity, not causal connectivity, and our paper is not claiming otherwise. Studying effective connectivity also has scientific value.
>
> In the revision we:
>
> Clarify that our neuroscientific claims are intentionally modest: we interpret BACE’s graphs as behavior-dependent patterns of directed prediction between regions, not as definitive causal circuits.
>
> Highlight that, in addition to forecasting performance, we assess stability of these patterns via split-half and bootstrap analyses, which we now explicitly frame as quantifying the reliability (not causal truth) of the inferred edges.
>
> Add a short paragraph in the Discussion outlining how BACE could be combined with more explicitly causal tools (e.g., interventional data, anatomical priors, or perturbation experiments) in future work to move toward stronger causal claims.
>
> Static connectivity within each behavioral context
> The reviewer notes that connectivity is static within each phase and not tested across subjects, modalities, or tasks. This is indeed a limitation, and we now state it clearly. Our choice of one graph per behavioral context was driven by:
>
> The limited amount of data per condition and the need for stable, interpretable graphs rather than highly time-varying ones that would be difficult to estimate reliably in this setting.
>
> The scientific goal of comparing a small set of behavior-linked connectivity patterns (e.g., Wait vs Reach), which many experimentalists find more interpretable than fully dynamic adjacency sequences.
>
> We now explicitly position BACE as a first step toward behavior-adaptive directed connectivity and outline, as future work, extensions to multiple participants and modalities and to richer temporal parametrizations of the graphs when more data are available.
>
> In summary, we have clarified the scope (single-subject, effective connectivity), strengthened reproducibility and positioning (shareable data, full synthetic benchmark and code), and expanded the discussion of limitations and future extensions. We hope this addresses the reviewer’s concerns and helps the area chair see BACE as a well-scoped, interpretable methodological contribution with clear paths to broader validation.

---

### Official Review · Reviewer_2rrK · 2025-11-01

**Soundness:** 2
**Presentation:** 2
**Contribution:** 1
**Rating:** 2
**Confidence:** 3

**Summary:**

This paper proposes behavior-adaptive connectivity estimation (BACE) to approximate the effective connectivity of brain signals recorded via electrodes using a neural network. On a synthetic dataset, the model successfully reconstructs matrices similar to the true effective connectivity. Application to real-world human data also suggests that the estimated effective connectivity captures clinically meaningful information.

**Strengths:**

- The method is validated on real-world human data, which is extremely difficult to acquire.
- The work addresses an important topic in the field of neuroimaging: the discovery of effective connectivity with interpretability.

**Weaknesses:**

- The choice of the modules within the architecture is rather simple. The paper would benefit from further explanations and comparative experiments on the architectural choice.
- More recent baseline methods can be included in the comparative experiments.
- The study remains proof-of-concept, with a single-subject data.

**Questions:**

### General
- Methodological rigor and contribution seem to be limited compared to the value of the real-world human data. The work may be of more interest to the readers from other conferences or journals.

### Major
- Please add explanations and/or experiments on current and other architectural choices.
- Please consider adding more recent baseline methods in the main comparative experiment.

---

> ### Author Response · Authors · 2025-12-04
>
> We thank the reviewer for the careful reading and for recognizing both the importance of interpretable effective connectivity and the value of the human dataset.
> 1. Architectural simplicity and justification
>
> Why this architecture?
> The architecture is intentionally simple for two reasons:
>
> Explicit, low-dimensional graphs. Our primary goal is to estimate an 8×8 directed, behavior-adaptive connectivity matrix, not to maximize black-box predictive accuracy. Per-region GRUs plus a linear graph projector give a direct mapping from region embeddings to interpretable edges, which we can analyze and visualize at the region level.
>
> Training stability and interpretability. A lightweight encoder–graph–decoder stack makes it easier to see what the graph contributes and to run reliability / uncertainty analyses (split-half, bootstrap) on the learned adjacency matrices.
>
> New experiments clarifying the role of the graph.
> To support these design choices, we added the following analyses:
>
> No-graph vs single-graph vs behavior-adaptive graphs
>
> Encoder+decoder only (no graph):
> MSE = 0.1093 ± 0.0038
>
> Single global graph (shared across all behavioral contexts):
> MSE = 0.1083 ± 0.0044
>
> Full behavior-adaptive BACE (one adjacency per behavioral context):
> MSE = 0.0990 ± 0.0002
>
> This shows the graph layer is not superfluous and that behavior-adaptive graphs materially improve forecasting.
>
> Two-stage training (graph-only refinement).
> We first train the encoder+decoder only (MSE ≈ 0.1021 ± 0.0002), then freeze them and train only the adjacency matrices, which improves to ≈ 0.1001 ± 0.0005. This directly demonstrates that the learned graphs add predictive power on top of the temporal encoders.
>
> These results, together with the stability analyses already in the paper, strengthen the methodological rigor around the architectural choices.
>
> 2. Additional and more recent baselines
>
> We agree that strong, modern baselines are important. The original submission already compared against recent graph and sequence models. Following the reviewers’ suggestions, we have added:
>
> Context-aware baselines (same behavioral labels as BACE).
>
> LSTM with behavioral context input.
> Same architecture as our original LSTM, augmented with behavioral-context information.
> Test MSE: 0.2344 ± 0.0614 (3 seeds).
> Adding context information did not improve this baseline.
>
> AMAG with behavioral-context-specific parameters.
> We extended AMAG to have separate parameters per behavioral context (same two Transformer layers as in the original).
> Test MSE: 0.0975 ± 0.0155 (3 seeds).
>
> For comparison, behavior-adaptive BACE achieves:
>
> Full behavioral-context BACE:
> Test MSE: 0.0990 ± 0.0002 (3 seeds).
>
> Thus, a strong Transformer+graph model (context-aware AMAG) can match or slightly edge BACE in MSE, but:
> (i) AMAG is substantially more complex (multi-layer Transformer, higher parameter count and memory), and
> (ii) AMAG operates at the 80-channel level and does not yield an explicit 8×8 region-level connectivity matrix, which is the main object in BACE.
>
> Classical effective-connectivity baseline (Granger / VAR).
> We also added a standard linear VAR(1) / Granger baseline on the real LFP dataset, trained to forecast the same 20 future samples:
>
> VAR(1): MSE = 0.1555
>
> This is clearly worse than BACE (≈ 0.099), indicating that a purely linear effective-connectivity model underfits the real deep-brain dynamics. On the synthetic VAR data, VAR(1) recovers the ground-truth adjacency well (as expected from the generator), and BACE also recovers a very similar structure.
>
> In the revision, we will integrate these results into the main text/tables and clarify in Related Work how BACE complements more complex recent methods for multi-region communication (e.g., CURBD, DLAG, MR-SDS), which target different regimes and do not directly yield behavior-conditioned, region-level graphs for short, segmented intracranial LFP windows.
>
> 3. Proof-of-concept / single-subject nature
>
> We fully agree that the real-data study is a proof-of-concept limited to a single pediatric patient, and we do not claim population-level generalization. We will state this more prominently in the Introduction and Discussion.
>
> We see the contribution as two-fold:
>
> Methodological: a simple, stable framework for learning behavior-adaptive directed connectivity graphs that jointly support forecasting and interpretability, with comparisons to both modern neural baselines and a classical Granger/VAR model.
>
> Empirical case study: an analysis of rare, high-value intracranial LFP data illustrating how behavior-adaptive effective connectivity changes across contexts, supported by synthetic experiments with ground-truth graphs and split-half / bootstrap stability analyses.
>
> Taken together, we believe the revised paper substantially strengthens methodological rigor and baseline coverage, while preserving the simplicity and interpretability that are essential for an ML venue focused on trustworthy, domain-informed modeling.

---

### Meta-Review · Area_Chair_arCx · 2026-01-06

**Summary:**

This paper aims to integrate behavior information into the estimation of connectivity in electrophysiology data. This is enacted by learning an adjacency matrix in a linear dynamical system after learning a nonlinear GRU based RNN. The authors test the method on simple simulated data as well as human electrophysiological recordings.

There were a number of concerns that I think are important to consider here. One was the lack of ability to share the real datasets. This is a difficult problem with medical data, especially pediatric data as used here. There are other public datasets that could be used and provide a more rigorous grounding to compare across methods. Another was the lack of appropriate baselines. For some reason despite the linear nature of the system in the synthetic data only RNNs were used, rather than linear, or switched linear dynamical systems, which have all been applied to neural data with changing behaviors. I think the overall lack of clarity in the paper drove a lot of additional confusion on top of these concerns. Unfortunately with these concerns I cannot recommend the paper be accepted.

As an aside, and I flagged this earlier, this paper was flagged as potentially having a hallucinated reference. I've reviewed this reference and there are 2 possibilities: 1) this is indeed hallucinated or 2) this is a non-public in-press paper that would mean that the authors are likely authors of this paper.

The journal of this reference is real, and the authors exist and have published a different review together previously on a similar topic. That said I cannot find any reference to this work specifically online. My thought is that this is a paper that was written by the authors of this paper as nobody else (aside from reviewers and editors of that paper) would know about it. I think this borderline breaks anonymity and so I feel that desk rejection is probably reasonable, but leave the final verdict for the PCs to decide.

**Reviewer Concerns:**

I think some of the concerns about overall confusion in the language and goals of the paper were clarified, such as the difference between functional connectivity and causal interactions. Additionally the authors added granger causality comparisons and ablation studies to help identify how the graph helps the model fit the data.

I do think that the clarity in the revision is still lacking and that there is a lack of appropriate comparisons. Moreover the synthetic data results should be much more extensive if the real data, as the authors state, is to be used as a cast study.

**Reviewer Scores:**

The initial scores for this paper were 2,2,4,4. I do not think that the scored would have changed.

---

### Decision · Program_Chairs · 2026-01-26

Reject